# Comprehensive, Comparative Evaluation of 35 Manual SARS-CoV-2 Serological Assays

Wayne Dimech,[a] Shannon Curley,[a] Lorenzo Subissi,[b] Ute Ströher,[b] Mark D. Perkins,[b] Jane Cunningham[b]

[a]National Serology Reference Laboratory, Fitzroy, Australia
[b]World Health Organization, Geneva, Switzerland

**ABSTRACT** The onset of the coronavirus disease 2019 (COVID-19) pandemic resulted in hundreds of *in vitro* diagnostic devices (IVDs) coming to market, facilitated by regulatory authorities allowing "emergency use" without a comprehensive evaluation of performance. The World Health Organization (WHO) released target product profiles (TPPs) specifying acceptable performance characteristics for severe acute respiratory syndrome coronavirus 2 (SARS-CoV-2) assay devices. We evaluated 26 rapid diagnostic tests and 9 enzyme immunoassays (EIAs) for anti-SARS-CoV-2, suitable for use in low- and middle-income countries (LMICs), against these TPPs and other performance characteristics. The sensitivity and specificity ranged from 60.1 to 100% and 56.0 to 100%, respectively. Five of 35 test kits reported no false reactivity for 55 samples with potentially cross-reacting substances. Six test kits reported no false reactivity for 35 samples containing interfering substances, and only one test reported no false reactivity with samples positive for other coronaviruses (not SARS-CoV-2). This study demonstrates that a comprehensive evaluation of the performance of test kits against defined specifications is essential for the selection of test kits, especially in a pandemic setting.

**IMPORTANCE** The markets have been flooded with hundreds of SARS-CoV-2 serology tests, and although there are many published reports on their performance, comparative reports are far fewer and tend to be limited to only a few tests. In this report, we comparatively assessed 35 rapid diagnostic tests or microtiter plate enzyme immunoassays (EIAs) using a large set of samples from individuals with a history of mild to moderate COVID-19, commensurate with the target population for serosurveillance, which included serum samples from individuals previously infected, at undetermined time periods, with other seasonal human coronaviruses, Middle East respiratory syndrome coronavirus (MERS-CoV), and SARS-CoV-1. The significant heterogeneity in their performances, with only a few tests meeting WHO target product profile performance requirements, highlights the importance of independent comparative assessments to inform the use and procurement of these tests for both diagnostics and epidemiological investigations.

**KEYWORDS** evaluation, SARS-CoV-2, serology

In November 2019, a novel acute respiratory disease (coronavirus disease 2019 [COVID-19]) caused by a new coronavirus (severe acute respiratory syndrome coronavirus 2 [SARS-CoV-2]) was first recognized. Since that time, a major global pandemic has ensued, causing significant mortality, morbidity, and economic disruption. Due to the high level of concern, regulators initially reduced their usual strict regulatory requirement that *in vitro* diagnostic device (IVD) manufacturers demonstrate evidence of adequate performance, safety, and quality. Most regulators allowed the use of IVDs under emergency use conditions, requiring limited premarket evidence of performance. Within 6 months of the start of the pandemic, more than 700 different SARS-CoV-2 IVDs were commercially available. Many of those first to market were antibody/serological rapid diagnostic tests (RDTs) and enzyme immunoassays (EIAs). Many early assay performance studies were poorly structured, used different

Address correspondence to Wayne Dimech, wayne@nrlquality.org.au.

The authors declare no conflict of interest.

target populations, had inappropriate interpretations, and/or assessed small numbers of test kits. Few findings were published in peer-reviewed journals (1–3). Due to these shortcomings and in the face of requests for guidance on appropriate use and procurement, the World Health Organization (WHO) published guidance (4) recommending that RDTs be for research use only and that "They should not be used in any other setting, including for clinical decision-making, until evidence supporting use for specific indications is available." The WHO also stated, "although research into their performance and potential diagnostic utility is highly encouraged" (4).

To address some of the shortcomings and to better shape guidance and inform the procurement of serological assays, the WHO commissioned the National Serology Reference Laboratory, Australia (NRL), a WHO collaborating center and authorized WHO IVD prequalification evaluation laboratory, to develop a comprehensive protocol and conduct a large comparative evaluation of both RDTs and laboratory-based anti-SARS-CoV-2 serology tests suitable for use in low- and middle-income countries (LMICs). An open call for expression of interest for manufacturers to participate in the evaluation scheme was issued (5). One hundred two products from 71 manufacturers were submitted, and 45 products from 44 manufacturers were accepted based on compliance with both entry and short-listing criteria. Nine of the short-listed manufacturers withdrew their applications, leaving 35 products. In July 2021, the NRL and the WHO began publishing summary results for all 35 products, describing key observations and implications for use for clinical care and surveillance.

The primary aims of this study were to produce a statistically significant assessment of the performances of test kits designed to detect different classes of antibodies to different SARS-CoV-2 antigens, directly compare performance data from a range of commercially available serology tests by using the same panel of samples, and provide comprehensive performance characteristics of serology assays and determine whether serology could serve any useful purpose in the diagnosis or surveillance of SARS-CoV-2 infections.

## RESULTS

The results of this study represent the performances of the versions of the product and lot numbers used, and other versions or lots may result in different findings. The performance of all IVDs should be monitored over time with a well-designed quality assurance program. The invalid test rate, sensitivity, and specificity results for RDTs and EIAs are presented in Figures 1 and 2.

**Invalid test rate.** The invalid test rate ranged from 0.00% to 1.40%. A total of 13 of the 23 RDTs that reported IgG and IgM had no invalid test results. Only Biogenix reported an invalid test rate of >1.00%. Of the 12 tests that reported single IgG, total antibody, or neutralizing (Nt) antibody results, 7 had no invalid results.

**Concordance with recent infection.** The results for testing samples positive for anti-SARS-CoV-2 are expressed as "percent concordance with recent infection," including 95% confidence interval (CI) ranges. The 23 RDTs were evaluated for reactivity to IgG, IgM, or IgG and/or IgM against SARS-CoV-2 (Fig. 1). Two products (AllTest G/M and Healgen) reported 100% concordance with recent infection for IgG. Three tests (Biocan, Deepblue, and Singclean) reported <90.0% concordance with recent infection for IgG. All other tests reported between 90.0 and 99.9% concordance with recent infection for IgG. None of the 23 RDT kits reported 100% concordance with recent infection for IgM, with 9/23 (39.1%) having between 90.0 and 99.9% concordance for IgM and the remaining 14/23 tests (60.9%) having <90.0% concordance with recent infection for IgM. Two kits, Lysun (55.8%) and Singclean (57.3%), reported the lowest percent concordances with recent infection for IgM.

**Specificity.** Twenty of the 23 RDTs (87.0%) had a specificity of >90.0% for IgG testing, with Getein (81.0%), Lysun (89.3%), and Sensing (56.0%) having specificities of <90.0%. Eleven of the 23 RDTs (47.8%) had specificities of >90.0% for IgM; 9 of 23 (39.1%) had specificities of between 70.0 and 90.0% for IgM; and 3, Dynamiker (34.3%), Sensing (48.0%), and Sugentech (68.3%), had specificities of <70.0%.

The percent concordances with recent infection and specificities of 12 test kits that

| Abbreviation | Invalid Rate | Percentage Concordance with Recent Infection [95% CI range] | | | Specificity [95% CI range] | | |
|---|---|---|---|---|---|---|---|
| | | IgG | IgM | IgG and/or IgM | IgG | IgM | IgG and IgM |
| AllTest G/M | 0.00% | 100.0 [97.6 - 100] | 90.0 [84.7 - 93.6] | 100.0 [97.6 - 100] | 94.0 [90.5 - 96.3] | 84.0 [79.2 - 87.9] | 80.7 [75.6 - 84.9] |
| Artron | 0.00% | 98.5 [95.3 - 99.6] | 97.0 [93.3 - 98.8] | 99.0 [96.0 - 99.8] | 96.7 [93.8 - 98.3] | 88.0 [83.7 - 91.3] | 85.7 [81.1 - 89.3] |
| Biocan | 0.63% | 89.4 [84.1 - 3.2] | 74.4 [67.6 - 80.2] | 89.4 [84.1 - 93.2] | 96.7 [93.8 - 98.3] | 96.0 [92.9 - 97.8] | 94.0 [90.5 - 96.3] |
| Biogenix | 1.40% | 94.5 [90.1 - 97.1] | 82.4 [76.2 - 87.3] | 96.0 [91.9 - 98.1] | 96.3 [93.3 - 98.1] | 82.3 [77.4 - 86.4] | 80.3 [75.3 - 84.6] |
| Biohit | 0.00% | 92.0 [87.0 - 95.2] | 77.4 [70.8 - 82.9] | 96.0 [91.9 - 98.1] | 99.0 [96.9 - 99.7] | 93.3 [89.7 - 95.8] | 92.7 [89.0 -95.2] |
| BioMedomics | 0.13% | 93.5 [88.8 - 96.3] | 79.4 [73.0 - 84.7] | 96.5 [92.6 - 98.5] | 97.3 [94.6 - 98.8] | 89.7 [85.5 - 92.8] | 87.7 [83.3 - 91.1] |
| Boson | 0.13% | 99.5 [96.8 -100] | 83.9 [77.9 - 88.6] | 100.0 [97.6 -100] | 97.7 [95.0 - 99.0] | 77.0 [71.7 - 81.6] | 75.7 [70.3-80.3] |
| BTNX | 0.00% | 94.0 [89.5 - 96.7] | 84.9 [79.0 - 89.4] | 97.5 [93.9 - 99.1] | 99.0 [96.9 - 99.7] | 97.3 [94.6 - 98.8] | 96.3 [93.3 - 98.1] |
| Core Tech | 0.00% | 99.5 [96.8 - 100] | 80.9 [74.6 - 86.0] | 99.5 [96.8 - 100] | 93.7 [90.1 - 96.0] | 90.3 [86.3 - 93.3] | 85.3 [80.7 - 89.0] |
| Deepblue | 0.13% | 77.4 [70.8 - 82.9] | 94.0 [89.5 - 96.7] | 96.5 [92.6 - 98.5] | 95.3 [92.1 - 97.3] | 70.0 [64.4 - 75.1] | 69.3 [63.7 - 74.4] |
| Dynamiker | 0.25% | 99.0 [96.0 - 99.8] | 99.0 [96.0 - 99.8] | 99.5 [96.8 - 100] | 96.0 [92.9 - 97.8] | 34.3 [29.0 - 40.0] | 34.0 [28.7 - 39.7] |
| Getein | 0.00% | 99.0 [96.0 - 99.8] | 79.4 [73.0 - 84.7] | 99.5 [96.8 - 100] | 81.0 [76.0 - 85.2] | 97.0 [94.2 - 98.5] | 79.3 [74.2 - 83.7] |
| Healgen | 0.00% | 100.0 [97.6 - 100] | 95.5 [91.3 - 97.8] | 100.0 [97.6 - 100] | 93.7 [90.1 - 96.0] | 90.3 [86.3 - 93.3] | 88.7 [84.4 - 91.9] |
| Joysbio | 0.00% | 99.0 [96.0 - 99.8] | 97.0 [93.2 - 98.8] | 99.5 [96.8 - 100] | 94.7 [91.3 - 96.8] | 96.0 [92.9 - 97.8] | 94.0 [90.5 - 96.3] |
| Lysun | 0.00% | 99.5 [96.8 - 100] | 55.8 [48.6 - 62.7] | 99.5 [96.8 - 100] | 89.3 [85.1 - 92.5] | 71.3 [65.8 - 76.3] | 65.0 [59.3 - 70.3] |
| MPBio | 0.76% | 99.5 [96.8 - 100] | 76.0 [69.2 - 81.5] | 99.5 [96.8 - 100] | 90.7 [86.7 - 93.6] | 83.7 [78.9 - 87.6] | 79.0 [73.9 - 83.4] |
| RightSign | 0.00% | 95.0 [90.7 - 97.4] | 80.4 [74.1 - 85.5] | 97.0 [93.2 - 98.8] | 98.3 [95.9 - 99.4] | 99.0 [96.9 - 99.7] | 97.7 [95.0 - 99.0] |
| Sensing | 0.13% | 99.5 [96.8 - 100] | 99.5 [96.8 - 100] | 100.0 [97.6 - 100] | 56.0 [50.2 - 61.7] | 48.0 [42.2 - 53.8] | 32.0 [26.8 - 37.6] |
| Singclean | 0.13% | 85.4 [79.6 - 89.9] | 57.3 [50.1 - 64.2] | 86.9 [81.3 - 91.1] | 99.0 [96.9 - 99.7] | 79.0 [73.9 - 83.4] | 78.7 [73.5 - 83.1] |
| Standard Q | 0.00% | 97.5 [93.9 - 99.1] | 75.4 [68.7 - 81.1] | 98.0 [94.6 - 99.4] | 98.3 [95.9 - 99.4] | 98.7 [96.4 - 99.6] | 97.7 [95.0 - 99.0] |
| Sugentech | 0.00% | 98.5 [95.3 - 99.6] | 97.0 [93.2 - 98.8] | 99.5 [96.8 - 100] | 97.7 [95.0 - 99.0] | 68.3 [62.7 - 73.5] | 66.7 [61.0 - 71.9] |
| Sure Status | 0.13% | 96.5 [92.6 - 98.5] | 72.9 [66.0 - 78.8] | 96.5 [92.6 - 98.5] | 98.3 [95.9 - 99.4] | 95.7 92.5 - 97.6] | 94.0 [90.5 - 96.3] |
| VivaDiag | 0.00% | 95.5 [91.3 - 97.8] | 92.0 [87.0 - 95.2] | 98.5 [95.3 - 99.6] | 98.7 [96.4 - 99.6] | 92.0 [88.2 - 94.7] | 91.0 [87.0 - 93.9] |

**FIG 1** Invalid test rates, concordances with recent infection (n = 199), and specificities (n = 300) of rapid test devices testing for both IgG and IgM. Concordance and specificity results are presented as a heat map, with shades of green representing results of >90%, shades of yellow representing results of between 60 and 90%, and orange representing results of <60%.

reported single results for IgG only, total antibodies, or neutralizing antibodies are presented in Fig. 2. AllTest G, OmniPath (IgG), and Wantai (total antibodies) reported 100% concordance with recent infection. Three of the 12 tests (25.0%) reported <80% concordance with recent infection, MDGen (60.1%), Omega (64.3%), and Serion (78.9%), all testing for IgG. Of the 12 tests reporting a single result for IgG or total antibodies, only MDGen reported 100% specificity. Epitope (74.0%) had the lowest specificity of the 12 test kits.

**Analytical sensitivity and lot-to-lot variation.** Three samples (COVID461, COVID491, and COVID492) were doubling diluted from 1:2 to 1:1,024 and were tested with two reagent/test lots of each of the 35 tests. The results of testing the doubling dilution series are presented in Table S1 in the supplemental material. There was a large range of analytical sensitivities reported by the different test kits. In several RDT kits, a nonreactive

| Abbreviation | Invalid Rate | Concordance with Recent Infection [95% CI range] | | Specificity [95% CI range] | |
|---|---|---|---|---|---|
| | | IgG | Total or Nt | IgG | Total or Nt |
| AllTest G | 0.00% | 100.0 [97.6 - 100] | NA | 87.7 [83.3 - 91.1] | NA |
| bioLytical | 0.50% | NA | 98.0 [94.6 - 99.4] | NA | 88.3 [84.0 - 91.6] |
| BioRad | 0.00% | NA | 86.4 [80.7 - 90.7] | NA | 97.7 [95.0 - 99.0] |
| Epitope | 0.25% | 92.5 [87.6 - 95.6] | NA | 74.0 [68.6 - 78.8] | NA |
| MDGen | NA | 60.1 [53.6 - 67.6] | NA | 100.0 [98.4 - 100] | NA |
| Omega | 0.00% | 64.3 [57.2 - 70.9] | NA | 85.3 [80.7 - 89.0] | NA |
| OmniPath | 0.00% | 100.0 [97.6 - 100] | NA | 98.0 [95.5 - 99.20] | NA |
| Serion | 0.00% | 78.9 [72.4 - 84.2] | NA | 99.3 [97.3 - 99.9] | NA |
| Vazyme | 0.00% | NA | 89.4 [84.1 - 93.2] | NA | 98.3 [95.9 - 99.4]. |
| Vircell | 0.00% | 93.5 [88.8 - 96.3] | NA | 92.0 [88.2 - 94.7] | NA |
| Wantai | 0.00% | NA | 100.0 [97.64 - 100] | NA | 99.7 [97.9 - 100] |
| Wondfo | 0.40% | NA | 98.5 [95.3 - 99.6] | NA | 97.0 [94.2 - 98.5] |

**FIG 2** Invalid test rates, concordances with recent infection, and specificities of rapid test devices that report a single result for IgG only, total antibodies, or neutralizing antibodies presented as a heat map, with shades of green representing results of >90%, shades of yellow representing results of between 60 and 90%, and orange representing results of <60%. NA, not assessed; Total, total antibodies; Nt, neutralizing antibodies.

result was followed by a reactive result with a higher dilution, making the interpretation of the results difficult.

All 35 test kits detected COVID461 at 1:2 for both IgG and IgM and COVID491 and COVID492 for IgG. However, some test kits did not detect IgM for COVID491 and/or COVID492 at 1:2. Dynamiker and Wantai reported reactive results for all 10 dilutions in some instances.

Four of the 35 tests reported a difference in the reactivities of two or more doubling dilutions when the same sample was tested with two different lots, including Lysun (IgG and IgM for COVID461), Singclean (IgM for COVID461), Biocan (IgM for COVID461), and Deepblue (IgG and IgM for COVID461 and -492).

**Cross-reactivity and interference.** The summary results for testing 55 samples containing potentially common cross-reacting analytes, 35 samples with interfering substances, and 31 samples from individuals with known past infection with SARS-CoV-1, Middle East respiratory syndrome coronavirus (MERS-CoV), and seasonal human coronavirus (HCoV) (HCoV-229E, HCoV-NL63, or HCoV-OC43) are presented Fig. 3, with the complete set of results shown in Table S2. There was a broad range in the numbers of false-reactive results for the cross-reacting, interfering, and non-SARS-CoV-2 coronavirus samples. Only five tests (MDGen, OmniPath, Serion, Standard Q, and Wantai) reported no false reactivity for the 55 samples with potentially cross-reacting substances. Of the 35 samples with potentially interfering substances, the 5 samples containing rheumatoid factor were falsely reactive by most tests. Dynamiker reported false IgM reactivity for 47 of 55 (85.5%) cross-reacting samples and 26 of 35 (74.3%) samples containing interfering substances. Six tests (Bio Hit, MDGen, OmniPath, Serion, Sure Status, and Wondfo) reported no false-reactive results for the 35 samples containing interfering substances.

MDGen was the only test kit that had no false reactivity across the cross-reacting, interfering, and non-SARS-CoV-2 panels. However, this test also failed to detect true-positive samples, reporting a low sensitivity of 60.1%.

| Abbreviation | Cross reactivity (n=55) | | | Interference (n=35) | | | Other Coronaviruses* (n=31) | | | SARS-CoV-1 (n=18) | | |
|---|---|---|---|---|---|---|---|---|---|---|---|---|
| | IgG | IgM | Total | IgG | IgM | Total | IgG | IgM | Total | IgG | IgM | Total |
| Alltest G | 4 | NA | NA | 3 | NA | NA | 14 | NA | NA | 12 | NA | NA |
| AllTest G/M | 3 | 5 | NA | 1 | 3 | NA | 2 | 3 | NA | 6 | 2 | NA |
| Artron | 1 | 7 | NA | 0 | 2 | NA | 10 | 1 | NA | 9 | 0 | NA |
| Biocan | 0 | 1 | NA | 0 | 1 | NA | 4 | 2 | NA | 3 | 1 | NA |
| Biogenix | 0 | 9 | NA | 3 | 7 | NA | 3 | 3 | NA | 2 | 1 | NA |
| Biohit | 0 | 2 | NA | 0 | 0 | NA | 14 | 3 | NA | 13 | 2 | NA |
| bioLytical | NA | NA | 11 | NA | NA | 5 | NA | NA | 12 | NA | NA | 10 |
| BioMedomics | 0 | 7 | NA | 2 | 9 | NA | 12 | 1 | NA | 12 | 0 | NA |
| BioRad | NA | NA | 2 | NA | NA | 1 | NA | NA | 15 | NA | NA | 14 |
| Boson | 1 | 8 | NA | 1 | 3 | NA | 16 | 3 | NA | 14 | 1 | NA |
| BTNX | 0 | 4 | NA | 3 | 7 | NA | 14 | 0 | NA | 14 | 0 | NA |
| Core Tech | 0 | 3 | NA | 2 | 0 | NA | 10 | 1 | NA | 9 | 0 | NA |
| Deepblue | 1 | 10 | NA | 7 | 8 | NA | 3 | 11 | NA | 1 | 6 | NA |
| Dynamiker | 3 | 47 | NA | 2 | 26 | NA | 11 | 17 | NA | 10 | 8 | NA |
| Epitope | 12 | NA | NA | 8 | NA | NA | 12 | NA | NA | 10 | NA | NA |
| Getein | 10 | 2 | NA | 12 | 1 | NA | 18 | 0 | NA | 18 | 0 | NA |
| Healgen | 1 | 7 | NA | 2 | 3 | NA | 17 | 6 | NA | 17 | 5 | NA |
| Joysbio | 5 | 4 | NA | 10 | 9 | NA | 16 | 10 | NA | 13 | 8 | NA |
| Lysun | 10 | 17 | NA | 7 | 15 | NA | 15 | 7 | NA | 11 | 3 | NA |
| MDGen | NA | NA | 0 | NA | NA | 0 | NA | NA | 0 | NA | NA | 0 |
| MPBio | 6 | 12 | NA | 5 | 7 | NA | 17 | 3 | NA | 14 | 0 | NA |
| Omega | NA | NA | 13 | NA | NA | 3 | NA | NA | 10 | NA | NA | 7 |
| OmniPath | 0 | NA | NA | 0 | NA | NA | 17 | NA | NA | NA | NA | 16 |
| RightSign | 0 | 1 | NA | 1 | 0 | NA | 19 | 1 | NA | 15 | 0 | NA |
| Sensing | 25 | 27 | NA | 21 | 19 | NA | 19 | 13 | NA | 17 | 9 | NA |
| Serion | 0 | NA | NA | 0 | NA | NA | 17 | NA | NA | 16 | NA | NA |
| Singclean | 0 | 29 | NA | 2 | 17 | NA | 4 | 22 | NA | 4 | 15 | NA |
| Standard Q | 0 | 0 | NA | 0 | 5 | NA | 9 | 1 | NA | 8 | 0 | NA |
| Sugentech | 2 | 19 | NA | 3 | 10 | NA | 1 | 4 | NA | 5 | 3 | NA |
| Sure Status | 0 | 3 | NA | 0 | 0 | NA | 6 | 2 | NA | 6 | 2 | NA |
| Vazyme | NA | NA | 1 | NA | NA | 2 | NA | NA | 17 | NA | NA | 16 |
| Vircell | 7 | NA | NA | 3 | NA | NA | 17 | NA | NA | 10 | NA | NA |
| VivaDiag | 0 | 7 | NA | 0 | 1 | NA | 6 | 2 | NA | 6 | 1 | NA |
| Wantai | NA | NA | 0 | NA | NA | 1 | NA | NA | 0 | NA | NA | 18 |
| Wondfo | NA | NA | 1 | NA | NA | 0 | NA | NA | 14 | NA | NA | 9 |

**FIG 3** Number of reactive results of testing samples from individuals having potentially cross-reacting or interfering substances or known past infection with SARS-CoV-1, MERS-CoV, and seasonal human coronavirus (HCoV-229E, HCoV-NL63, or HCoV-OC43). A heat map presents test kit results, with those having fewer than 10 false-reactive results for each population highlighted in shades of green. Those with between 10 and 30 false-reactive results are highlighted in shades of yellow, and those with >30 false-reactive results are highlighted in orange. NA, not applicable. *Includes samples from individuals having past MERS-CoV and seasonal coronavirus infections and excludes SARS-CoV-1 samples.

**Late-seroconversion panels.** Most kits reported results reactive for the analyte tested (IgG/IgM/total/Nt) for all serial blood samples. The IgM results from BTNX, Deepblue, Dynamiker, Healgen, Lysun, Sensing, Standard Q, Sugentech, and VivaDiag all reported one or more negative IgM results about 30 to 40 days after symptom onset. For one five-member series, four test kits (BioHit, Biomedomics, Getein, and RightSign) did not detect IgM in any of the samples.

**Seroconversion panels.** The results for seroconversion panels are summarized in Table S4. Samples were drawn from 8 days before to up to 52 days after the start of symptoms. Most test kits detected SARS-CoV-2 IgG and IgM antibodies within the first

**TABLE 1** Repeatability and reproducibility results, expressed as percent coefficients of variation, of enzyme immunoassays reporting quantitative results[a]

| | %CV | |
|---|---|---|
| EIA | Repeatability | Reproducibility |
| Bio-Rad | 7.80 | 10.40 |
| Epitope | 5.98 | 6.45 |
| MDGen | 11.37 | 4.69 |
| Omega | 6.26 | 13.42 |
| OmniPath | 3.50 | 5.60 |
| Serion | 4.79 | 11.99 |
| Vazyme | 3.70 | 3.52 |
| Vircell | 7.38 | 15.29 |
| Wantai | 8.82 | 15.50 |

[a]%CV, percent coefficient of variation.

week postinfection. Generally, the IgM response was detected earlier than or at the same time as the IgG response. There were some notable exceptions. MDGen, testing for IgG only, failed to detect antibodies in one patient's series of 9 samples and reported 5 negative results, 6 equivocal results, and 1 reactive result for a second patient's series of 14 samples. Serion, also testing for IgG only, reported negative results for the first five of a series of nine samples. The IgM responses by both RightSign and Standard Q decreased to undetectable levels in the same two of five seroconversion panels.

**Repeatability and reproducibility.** Repeatability and reproducibility studies were conducted on six EIAs. The results were expressed as the percent coefficient of variation (CV%) and are summarized in Table 1. Repeatability ranged from 3.70% to 11.37%, and reproducibility ranged from 3.52% to 13.42%.

## DISCUSSION

Within 6 months of the start of the pandemic, numerous antibody detection SARS-CoV-2 RDTs and EIAs became available. Regulators allowed the use of these novel tests through some form of emergency use authorization, requiring manufacturers to provide only limited evidence of test kit performance. Numerous studies comparing the performances of test kits were published (2, 3, 6, 7), often as preprints (8), which were not subject to rigorous peer review (2, 7). Early in the pandemic, the utility of SARS-CoV-2 serology testing was unknown but was used due to the lack of inexpensive point-of-care testing options and the long turnaround times for molecular diagnostics (7, 9, 10). RDTs claiming to detect IgM were used to diagnose recent infections (11, 12). To develop rational guidance on use and inform procurement, a comprehensive, head-to-head comparison of test kits was established, and the results were compared with published WHO target product profiles (TPPs) specifying acceptable and desirable performance characteristics, with RDT sensitivity being acceptable at ≥90% and desirable at ≥95% and specificity being acceptable at ≥97% and desirable at ≥99% and higher-throughput assays having acceptable and desirable sensitivities of ≥95% and >98% and specificities of ≥97% and ≥99%, respectively (13).

This study used samples from individuals with a recent history of mild-to-moderate clinical disease. The percent concordances of the results of IgM assays compared with nucleic acid amplification testing (NAT)-confirmed recent infection and specificity were highly variable, ranging from 55.8 to 99.5% and 34.3 to 99.0%, respectively. Five of the 23 test kits (27.3%) that reported IgM results reported more than 30 of the 90 cross-reacting and interfering substance-containing samples as being falsely reactive. There is some evidence that the IgM response decreases over time. No test achieved both the acceptable sensitivity and specificity criteria of TPPs based on the detection of IgM. These findings support the position that there is very limited clinical and epidemiological utility of IgM antibody testing (14).

All tests evaluated detected SARS-CoV-2 IgG either independently (IgG only), in association with the detection of IgM (IgG/IgM), or as a total antibody (IgG, IgM, and IgA) or neutralizing antibody test. Eight RDTs reporting IgG achieved acceptable levels

of both sensitivity and specificity (Fig. 3). No RDT had desirable levels of both sensitivity and specificity. EIAs are the preferred method to assess seroprevalence (2, 15). Wondfo and OmniPath met the acceptable TPP criteria and Wantai met the desirable criteria for both sensitivity and specificity, respectively (Fig. 2). These findings support the use of these limited numbers of RDT and EIA products for serosurveillance or retrospective diagnosis (for unvaccinated individuals).

In addition to some products achieving high levels of concordance with recent infection and specificity, some also had very low reactivity with cross-reacting substances. More specifically, Wantai (EIA) reported just one false-reactive result from the 35 samples with interfering substances and none from the 55 cross-reacting samples, whereas Wondfo (RDT) reported no and one false-reactive result, respectively. bioLytica (RDT) and Bio-Rad (EIA) reported 5/35 and 11/55 and 1/35 and 2/55 false-reactive results, respectively.

The %CVs for the repeatability of seven tests that reported quantitative results ranged from 3.70 to 11.37%, whereas the %CVs for reproducibility ranged from 3.52 to 15.50%. The imprecision of quantitative tests should continually be monitored using a well-developed quality control (QC) program (16).

The results of this study indicate that the SARS-CoV-2 IgG response is detectable at the same time as or one bleed after the detection of IgM (10). The results of the seroconversion and late-seroconversion panels indicate that there was little evidence that the IgG response became undetectable within 7 weeks after infection, which is consistent with the results of other studies (10, 17).

This study demonstrated that some tests had unacceptably poor concordance with recent infection and specificity, and others reported unacceptably high levels of false reactivity due to cross-reactive and interfering substances and antibodies from other coronaviruses. Most tests were reactive within the first week after the onset of symptoms. Several tests demonstrated a >2-fold difference in analytical sensitivity between the two lots, indicating inconsistent manufacturing practices.

Several limitations of this study should be noted. This study did not assess safety, usability, cost, or test kit robustness. This study used predominantly citrated plasma samples collected using plasmapheresis. This study, along with many others, applied tests in laboratory settings on plasma or serum samples, while they are also approved for use as point-of-care tests using (capillary) whole blood; therefore, it is not possible to ascertain the clinical accuracy of these tests in the intended settings of use. Some studies suggest a performance comparable to that with whole blood (18, 19).

All positive samples were from individuals infected with the ancestral variant. This study did not evaluate the test kits using samples obtained from individuals vaccinated with varying vaccines or numbers of doses, with or without a history of infection. Additional studies in these populations will be required. Some of the samples used in the cross-reacting and interfering substance panels had limited clinical information.

The landscape of clinical diagnostics for SARS-CoV-2 has drastically evolved since the initial antibody tests emerged on the scene; point-of-care SARS-CoV-2 antigen- and molecular-based tests and expanded PCR laboratory capacities now fill the acute diagnostic needs (13). This evaluation was an attempt to better inform the role of antibody testing and subsequent procurement as part of the pandemic response. In the future, mechanisms should be on standby to allow more rapid comparative evaluations to identify good- and poor-performing products and better understand the appropriate use. Although this was a large study of 35 products, it included only a fraction of the products on the market and revealed dramatic variability in performance across various parameters. Nonetheless, a small number of products met WHO TPP criteria and, in the right context, could play a useful role in serosurveillance and epidemiological research. These results, coupled with more stringent regulatory requirements, may be useful in selecting products for these purposes.

## MATERIALS AND METHODS

**Test kit selection.** In November 2020, the WHO issued an expression of interest and the evaluation protocol (5). The following exclusion criteria were used: products targeting IgM or IgA only; products

**TABLE 2** Final list of test kits included in the WHO SARS-CoV-2 serology evaluations, including abbreviations used in this report[a]

| Test kit abbreviation | Manufacturer | Product name | Product code | IFU version (date [yr/mo/day or mo/yr]) | Test type | Target |
|---|---|---|---|---|---|---|
| AllTest G | Hangzhou AllTest Biotech Co. Ltd. | COVID-19 IgG rapid test | INCPG-402 | 14253502 (2020/11/7) | RDT | IgG |
| AllTest G/M | Hangzhou AllTest Biotech Co. Ltd. | 2019-nCoV IgG/IgM rapid test cassette | INCP-402B | 14347400 | RDT | IgG/IgM |
| Artron | Artron Laboratories Inc. | Artron COVID-19 IgM/IgG antibody test | A03-51-322 | A03-51-322 version 09 (1/2021) | RDT | IgG/IgM |
| Biocan | Biocan Diagnostics Inc. | Biocan novel coronavirus (COVID-19) IgG/IgM antibody test | B251C | B251C (7/2020) | RDT | IgG/IgM |
| Biogenix | Biogenix Inc. Pvt. Ltd. | SARS CoV-2 IgM/IgG rapid test | Not provided | BIPL-IFU-081 | RDT | IgG/IgM |
| Biohit | Biohit Healthcare (Hefei) Co. Ltd. | Biohit SARS-CoV-2 IgM/IgG antibody test kit | 207.01.25.02 | Version 03 (2020/6/16) | RDT | IgG/IgM |
| bioLytical | bioLytical Laboratories Inc. | Insti COVID-19 antibody test | 90-1098 | 51-1311F (2021/11/3) | RDT | Total Ab |
| BioMedomics | BioMedomics Inc. | COVID-19 IgM-IgG rapid test | Not provided | 51-002 51-PI-002.CE (Rev 04) (2020/12/16) | RDT | IgG/IgM |
| Bio-Rad | Bio-Rad | Platelia SARS-CoV-2 total Ab | 12013798 | 16008267 (6/2020) | EIA | Total Ab |
| Boson | Xiamen Boson Biotech Co. Ltd. | Rapid 2019-nCoV IgG/IgM combo test card | 1N38C2 | 081985/200612 | RDT | IgG/IgM |
| BTNX | BTNX Inc. | Rapid-response COVID-19 IgG/IgM rapid test device | COV-13C25 | 1110032621 Rev 5.2 (2021/2/11) | RDT | IgG/IgM |
| Core Tech | Core Technology Co. Ltd. | Coretests COVID-19 IgM/IgG Ab test | Not provided | COVID IgM/IgG/05-C version 1.3 (9/2020) | RDT | IgG/IgM |
| Deepblue | Anhui Deepblue Medical Technology Co. Ltd. | COVID-19 (SARS-CoV-2) IgG/IgM antibody test kit (colloidal gold) | Not provided | COVID IgG/IgM-01 version 1.8 | RDT | IgG/IgM |
| Dynamiker | Dynamiker Biotechnology (Tianjin) Co. Ltd. | 2019-nCoV IgG/IgM rapid test | DNK-1419-1 | CE-SYSM-008 1.1 (1/2021) | RDT | IgG/IgM |
| Epitope | Epitope Diagnostics Inc. | EDI novel coronavirus COVID-19 IgG ELISA | KT-1032 | Version 13 (5/2021) | EIA | IgG |
| Getein | Getein Biotech Inc. | One-step test for novel coronavirus (2019-nCoV) IgM/IgG antibody (colloidal gold) | CG2057 | WCG76-SIN-DX-S-01 | RDT | IgG/IgM |
| Healgen | Healgen Scientific Limited Liability Company | COVID-19 IgG/IgM rapid test cassette | GCCOV-402a | B21901-01 (2020/6/5) | RDT | IgG/IgM |
| Joysbio | Joysbio (Tianjin) Biotechnology Co. Ltd. | Joysbio COVID-19 IgG/IgM rapid test kit | Not provided | V.1.0 (8/2020) | RDT | IgG/IgM |
| Lysun | Hangzhou Lysun Biotechnology Co. Ltd. | 2019-nCoV IgG/IgM rapid test device (colloidal gold) | COV-102 | 01 (5/2020) | RDT | IgG/IgM |
| MDGen | MicroDigital Co. Ltd. | MDGen AB96-COVID-19 IgG | Not provided | 1.0.3 | EIA | IgG |
| MPBio | MP Biomedicals Asia Pacific Pte. Ltd. | VivaDiag SARS-CoV-2 IgM/IgG rapid test | 43140-020 | MDC0011-ENG-3 (11/2020) | RDT | IgG/IgM |
| Omega | Genesis Diagnostics Ltd. (subsidiary of Omega Diagnostics Group PLC) | Omega Diagnostics COVID-19 IgG ELISA kit | ODL150/10 | 263-ODL 150/10 version 6.0 (2/2021) | EIA | IgG |

**TABLE 2** (Continued)

| Test kit abbreviation | Manufacturer | Product name | Product code | IFU version (date [yr/mo/day or mo/yr]) | Test type | Target |
|---|---|---|---|---|---|---|
| OmniPath | Thermo Fisher Diagnostics | OmniPath 96 Combi SARS-CoV-2 IgG ELISA kit | R250120 | IFU X9487A, revised December 2020 | EIA | IgG (quantitative) |
| RightSign | Hangzhou Biotest Biotech Co. Ltd. | RightSign COVID-19 IgG/IgM rapid test cassette | INGM-MC42 | RP5381700 (2021/3/5) | RDT | IgG/IgM |
| Sensing | Sensing Self Pte. Ltd. | COVID-19 rapid IgG/IgM combined antibody assay prescreening test kit | ERCSS05310 | Version 1.1 (2021/2/19) | RDT | IgG/IgM |
| Serion | Institut Virion/Serion GmbH | Serion ELISA agile SARS-CoV-2 IgG | ESR400G | V a400AG-1 (9/2020) | EIA | IgG |
| Singclean | Hangzhou Singclean Medical Products Co. Ltd. | COVID-19 IgG/IgM test kit (colloidal gold method) | Not provided | A/0 (2020/8/3) | RDT | IgG/IgM |
| Standard Q | SD Biosensor Inc. | Standard Q COVID-19 IgM/IgG Plus test | Q-NCOV-02C | L23COV7DMENRO (2/2021) | RDT | IgG/IgM |
| Sugentech | Sugentech Inc. | SGTi-flex COVID-19 IgM/IgG | COVT025E | IS209E-05 (2020/9/1) | RDT | IgG/IgM |
| Sure Status | Premier Medical Corporation Private Limited | Sure Status COVID-19 IgG/IgM card test | SS02P25 | SS02-INS-001 (2020/11/10) | RDT | IgG/IgM |
| Vazyme | Nanjing Vazyme Medical Technology Co. Ltd. | Anti-SARS-CoV-2 neutralizing antibody ELISA | C8909C | 1 February 2021 | EIA | Nt Ab |
| Vircell | Vircell SL | COVID-19 ELISA IgG | G1032 | L-G1032-EN-05 (2021/2/23) | EIA | IgG |
| VivaDiag | VivaChek Biotech (Hangzhou) Co. Ltd. | VivaDiag SARS-CoV-2 IgM/IgG rapid test | VID35-08-011 | 1604003703 (2020/4/20) | RDT | IgG/IgM |
| Wantai | Beijing Wantai Biological Pharmacy Enterprise Co. Ltd. | Wantai SARS-CoV-2 Ab ELISA | WS-1096 | V.2020-02 (2020/6/22) | EIA | Total Ab |
| Wondfo | Guangzhou Wondfo Biotech Co. Ltd. | 2019-nCoV antibody test (lateral flow method) | W195P0004 | A2 (2020/12/8) | RDT | Total Ab |

aIFU, instructions for use; RDT, rapid diagnostic test; EIA, enzyme immunoassay; Nt Ab, neutralizing antibody; ELISA, enzyme-linked immunosorbent assay.

**TABLE 3** Cross-reacting panel comprising 55 samples containing common cross-reacting analytes and a subset of 31 samples containing SARS-CoV-1, MERS-CoV, or seasonal HCoV antibodies[a]

| Analyte | No. of samples |
| --- | --- |
| CMV IgM positive | 4 |
| EBV VCA IgM positive[b] | 2 |
| Influenza A virus positive | 3 |
| Influenza B virus positive | 3 |
| Hepatitis A virus IgM positive | 1 |
| Hepatitis B virus e antigen positive | 3 |
| Hepatitis B virus surface antigen | 5 |
| Hepatitis B virus surface antigen/hepatitis B virus c IgM positive | 1 |
| Hepatitis B virus surface antigen/hepatitis B virus c IgM/hepatitis B virus e antigen positive | 1 |
| Hepatitis C virus antibody positive | 4 |
| HIV antibody positive | 8 |
| Malaria antibody positive | 5 |
| Mycoplasma IgM positive | 1 |
| Parainfluenza virus positive | 1 |
| Parvovirus B19 IgM positive | 2 |
| Chlamydia psittaci IgM positive | 1 |
| Rubella virus IgM positive | 1 |
| Syphilis positive | 6 |
| Toxoplasma IgM positive | 3 |
| Severe acute respiratory syndrome coronavirus | 18 |
| Middle East respiratory syndrome coronavirus | 4 |
| Human seasonal coronavirus (HCoV-229E, HCoV-NL63, or HCoV-OC43) | 9 |

[a]CMV, cytomegalovirus; EBV, Epstein-Barr virus.
[b]VCA, viral capsid antigen.

needing proprietary platforms; products for which kit instructions for use (IFUs) were not included in the application; manufacturers without a free-sales certificate or ISO13485 accreditation; products that had low accuracy in early evaluations performed by the Foundation for New and Innovative Diagnostics (FIND) (6) (low accuracy defined as <80% sensitivity and <98% specificity); RDTs targeting anti-N antibodies only; and multiple products from a single manufacturer, with the exception of EIAs targeting anti-N antibodies.

The latter two criteria were adopted considering the potential for future seroprevalence studies after mass vaccination campaigns with spike protein-based vaccines. Furthermore, RDTs targeting anti-N antibodies alone were excluded because the literature indicated that anti-N antibody titers decay more quickly than anti-S antibodies and therefore may be less sensitive for the detection of past infection (20). Serosurveillance would require high-accuracy, high-throughput assays such as EIAs (2). Of the test kits selected, 26 were RDTs, and 8 were EIAs. One EIA (OmniPath) was added at a later stage as it was the commercialized version of the product used in the RECOVERY trial, the data from which suggested that the use of serology could help select those patients who were most likely to benefit from treatment with a monoclonal antibody cocktail (Regeneron) (21). The complete list of test kits evaluated is summarized in Table 2. All selected test kits were provided to the NRL free of charge.

**Sample panels.** The performance characteristics evaluated depended on the class(es) of antibodies being detected and the method for the reporting of results. All test kits were evaluated for sensitivity (concordance with documented SARS-CoV-2 RNA positivity by quantitative PCR [qPCR]), specificity, analytical sensitivity, quantification, lot-to-lot variation, seroconversion, cross-reactivity, and interference.

Test kits reporting quantitative results (e.g., sample-to-cutoff [S/Co] values) were evaluated for repeatability and reproducibility.

Samples contained various anticoagulants, including sodium citrate or citrate dextrose solutions. The anticoagulants used in some other samples were unknown. Some test kits evaluated specified the use of certain anticoagulants in the IFU. False reactivity due to the anticoagulants used in the panel cannot be discounted, and the results should be interpreted accordingly.

**(i) Sensitivity/concordance with recently confirmed SARS-CoV-2 infection.** A total of 199 samples were obtained by two commercial organizations (BioMex, Heidelberg, Germany [BioMex], and Medical Research Networx Biologicals, FL, USA [MRN]) from nonhospitalized individuals with a recent history of clinical infection with ancestral SARS-CoV-2, confirmed by various commercial NATs. As samples were collected from individuals between January and April 2020, it is assumed that infections were not due to Delta or Omicron variants. These samples were collected between 14 and 71 days after the onset of symptoms or after a positive NAT result. The results of the positive sample panels were reported as "concordance with recent infection." Approximately half of the panel was tested by each of two reagent/test lots.

**(ii) Specificity.** A total of 300 plasma samples obtained from NRL's sample bank, collected prior to November 2019, were used as the specificity panel. These samples were obtained from healthy blood donors and screened negative for blood-borne infections by serology and NATs. These samples were

**TABLE 4** Results of subjectively read assays

| Scoring index | Intensity reading scale |
| --- | --- |
| 0 | Nonreactive |
| 1 | Very weak |
| 2 | Weak |
| 3 | Medium-to-strong reactivity |

assumed to be negative for SARS-CoV-2 antibodies, and no further confirmation testing was performed. Approximately half of the panel was tested by each of two reagent/test lots.

**(iii) Analytical sensitivity/lot-to-lot variation.** Three of the sensitivity panel samples had 10 doubling dilutions, from 1:2 to 1:1,024, prepared in human plasma negative for SARS-CoV-2 antibodies. All dilutions were tested by two reagent lots.

**(iv) Cross-reactivity.** A total of 55 plasma or serum samples known to contain potentially cross-reacting analytes were tested by a single reagent/test lot along with a further 31 samples confirmed to be positive by NATs for severe acute respiratory syndrome coronavirus (SARS-CoV-1), Middle East respiratory syndrome coronavirus (MERS-CoV), or seasonal human coronavirus (HCoV-229E, HCoV-NL63, or HCoV-OC43) (Table 3). Samples were obtained from individuals with evidence of past infection with the organism indicated, unless specifically indicated by IgM reactivity.

**(v) Interfering substances.** A total of 35 plasma samples known to contain potentially interfering substances were tested by a single reagent/test lot. The interfering substance panel consisted of 5 visibly icteric samples, 5 visibly hemolyzed samples, 7 samples with visibly high levels of bilirubin, 5 lipemic samples, 5 samples with antinuclear antibodies, 3 samples positive for antibodies to double-stranded DNA (lupus), and 5 samples positive for rheumatoid factor.

**(vi) Late-seroconversion panels.** The late-seroconversion panel was comprised of 47 plasma samples collected by BioMex from 10 different, nonhospitalized, volunteer donors at various intervals commencing from 18 days or later after symptom onset. The purpose of this panel was to demonstrate the decline in IgM antibody titers over time.

**(vii) Seroconversion panels.** Seroconversion panels consisted of a total of 60 plasma samples collected by MRN from five different SARS-CoV-2 NAT-positive individuals at regular intervals from early infection to approximately 8 weeks after symptoms. The results of testing were used to determine the number of days after the onset of symptoms when the test kit first detected reactivity.

**(viii) Repeatability.** For repeatability studies (within-run precision), a positive sample diluted in negative plasma to give a low positive reaction or a commercial anti-SARS-CoV-2 quality control (QC) sample (DiaMex, Heidelberg, Germany) was tested 30 times in the same test run. The percent coefficient of variation (%CV) was calculated.

**(ix) Reproducibility.** For reproducibility studies, the same sample used in the repeatability study was tested 30 times across no fewer than five different runs, and the results were presented as the %CV.

**Testing protocol. (i) Rapid diagnostic tests.** Rapid diagnostic testing was performed according to the test IFU by one operator. The results were read by that operator and independently read by a second reader. The intensities of the test and control lines were graded according to a defined scale (Table 4). When consensus for the sample reading was not met, a third, independent reader recorded their result, and the eventual consensus (2 of 3 readings being the same) was used as the final result. The number of invalid results, as defined by the IFU, was recorded.

**(ii) Enzyme immunoassays.** Enzyme immunoassays were performed singly according to the IFU by the same operator. Invalid test runs were defined as when the kit controls failed the manufacturer's validation criteria.

All results, recorded on hard-copy result sheets at the time of reading and manually transcribed into Microsoft Excel, were double-checked by a second, independent person daily.

## SUPPLEMENTAL MATERIAL

Supplemental material is available online only.
**SUPPLEMENTAL FILE 1**, XLSX file, 0.03 MB.
**SUPPLEMENTAL FILE 2**, XLSX file, 0.04 MB.
**SUPPLEMENTAL FILE 3**, XLSX file, 0.02 MB.
**SUPPLEMENTAL FILE 4**, XLSX file, 0.03 MB.

## ACKNOWLEDGMENTS

We thank NRL scientific staff, including Sadaf Mohiuddin, Jingjing Cai, Bethmi Liyanage, and all technical support staff, who took part in composing the panels and performing testing. In particular, we acknowledge Technopath Clinical Diagnostics (Ballina, Ireland) for the generous donation of positive donor samples, including the seroconversion samples. Other panel samples were obtained commercially from Boca Biolisitics (FL, USA), BioMex (Heidelberg, Germany), Medical Research Networx (FL, USA),

and Seracare (MA, USA). We also acknowledge the Duke-NUS Medical School, the Erasmus Medical Center, Tan Tock Seng Hospital, and the International Vaccine Institute (IVI) for contributing samples from individuals previously infected with SARS-CoV-1, MERS-CoV, and seasonal human coronaviruses.

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
