## [Reviewer comments · Microbiology Spectrum]

Microbiology Spectrum

Comprehensive, Comparative Evaluation of 35 Manual SARS-CoV-2 Serological Assays

Wayne Dimech, Shannon Curley, Lorenzo Subissi, Ute Ströher, Mark Perkins, and Jane Cunningham

Corresponding Author(s): Wayne Dimech, National Reference Laboratory, Australia

Review Timeline:

Submission Date:	December 12, 2022
Editorial Decision:	March 22, 2023
Revision Received:	April 8, 2023
Accepted:	April 14, 2023

Editor: Oliver Laeyendecker

Reviewer(s): The reviewers have opted to remain anonymous.

Transaction Report:

DOI: <https://doi.org/10.1128/spectrum.05101-22>

March 22, 2023

Dr. Wayne Dimech
National Reference Laboratory, Australia
4th Floor Healy Building
41 Victoria Parade
Fitzroy, Victoria 3065
Australia

Re: Spectrum05101-22 (Comprehensive, Comparative Evaluation of 35 Manual SARS-CoV-2 Serological Assays)

Dear Dr. Wayne Dimech:

Thank you for submitting your manuscript to Microbiology Spectrum. As you will see your paper is very close to acceptance. Please modify the manuscript along the lines I have recommended. As these revisions are quite minor, I expect that you should be able to turn in the revised paper in less than 30 days, if not sooner. If your manuscript was reviewed, you will find the reviewers' comments below.

When submitting the revised version of your paper, please provide (1) point-by-point responses to the issues raised by the reviewers as file type "Response to Reviewers," not in your cover letter, and (2) a PDF file that indicates the changes from the original submission (by highlighting or underlining the changes) as file type "Marked Up Manuscript - For Review Only". Please use this link to submit your revised manuscript. Detailed instructions on submitting your revised paper are below.

Link Not Available

Sincerely,

Oliver Laeyendecker

Reviewer comments:

Reviewer #1 (Comments for the Author):

In this manuscript Dimech et al. perform a qualitative and comparative evaluation of a number of rapid tests (35) used for the detection of SARS-CoV-2 infection. This included 26 rapid diagnostic tests and 8 immunoassays. The manuscript is well written and the data clear. The authors performed an extensive and well-controlled study in which they tested samples expected to be positive, samples expected to be negative and then a number of other samples to account for cross-reactivity and interference. Most of the tests evaluated in this study were actually not performing optimally and did not fulfill the WHO requirements. The strengths of the manuscript rely on the number of tests included in the study, the well-controlled and stringent conditions used, and that the manuscript is also well-balanced thus acknowledging that the study does not evaluate samples from vaccinees or with a past of prior infection. I found the data of value. The study shows how difficult it is to develop a reliable test with great sensitivity and specificity. However, as a main weakness it would have been desirable that the study was done earlier since at this stage of the pandemic it is not so well-timed.

I only have some minor comments:

Line 42: I suggest that the authors clarify here if the individuals were convalescent or currently infected when the samples were taken.

Line 37: I suggest a different use of the comma, please revise.

Line 69, 71 and 89: There seem to be links here that are not working, or perhaps it is just the text that was made blue and underlined during the writing process, please correct.

Line 136: please define NAT at first use for enhanced clarity.

Line 161: please clarify the difference between icteric samples and the ones with high levels of bilirubin in line 163.

Table S2: what is A/B?

Supplementary materials: I think it would be helpful to add a legend or the like at the bottom of the data.

Line 261: out of curiosity, do the authors have a hypothesis or explanation for the samples with rheumatoid factor being the most falsely reactive? Would it be the increased number of autoantibodies?

Some typos:

Line 132: there's a parenthesis left behind.

Line 342: test should read tests.

Along the text: I would add a hyphen in Hepatitis B e-antigen for clarity.

Line 387: wouldn't RDT be rapid diagnostic test and not device?

Reviewer #2 (Comments for the Author):

The assessment of performance is sound. However, the authors need to clearly specify in the abstract (and body of the manuscript) that they are reviewing the performance of "rapid tests" for SARS-CoV-2 antibodies. It is not clear that this is the case until you start reading the body of the manuscript. Rapid tests are pretty much all some variation of disposable lateral flow immunoassay cartridge (or card, or cassette) and hence are all qualitative measures or semi-quantitative at best (although semi-quantitative claims for lateral flow assays are dubious). It would behoove the authors to specify the format of these rapid tests in such a way, i.e., refer to them as lateral flow immunoassays, which is a much more accurate description than calling them ELISAs, which conjures up the thought of a microtiter well plate with the reacting antibodies being read by a spectrophotometer and quantitated against a calibration curve. If there are any rapid assays that deviate from the lateral flow cartridge format, then state that as well, but in my quick review of the methods assessed they all sound like lateral flow immunoassays, or some variation of that.

2.5 years from the start of the pandemic in developed parts of the world use of these rapid assays for clinical purposes is not really happening. Developed countries are, for the most part, all using semi-quantitative or quantitative immunoassays for antibody measurement that require large-scale immunoassay instruments to perform. Thus, it would be appropriate to specify where (what countries) these rapid tests are still being used in, their prevalence of use, and for what purpose (i.e., clinical, research or both). There are some grammatical issues and typos to address (e.g., line 33 "defines" should be "defined", ELISA" is first used on line 40 without definition of the acronym, etc.)

It should also be reinforced throughout the manuscript that these are rapid assays that are being analyzed, i.e., rather than referring to these tests as "serology IVDs" or "IVDs" (e.g., line 29, 30, 81, change to "rapid antibody tests". Additionally, the authors seem to separate these rapid tests into two categories, "rapid diagnostic test" or "enzyme immunoassay", but what really matters is whether they were all qualitative assays that were run on disposable cartridges; if so, I would define those as all falling into a "rapid test" category. Again, what constitutes a "rapid test" needs to be specified from the outset, and again, as far as I can tell they are all rapid assays. Readers outside laboratory medicine do not understand the differences in assay formats and may conflate this as all antibody IVDs perform similarly to rapid tests. Other antibody IVDs for SARS-CoV-2 being used clinically are CMIA, ECLIA or similar format run on platforms like Abbott, Roche, Diasorin, MSD, Ortho, Siemens, etc., all of which provide semi-quantitative and quantitative measures and are far more robust for other reasons as well (e.g., sensitivity, specificity, etc.). Therefore, I cannot emphasize the importance of clearly defining what types of antibody tests we are talking about, and if these are all in fact rapid qualitative lateral flow immunoassays (or some variation of this), then they should definitely not be conflated with the semi-quantitative and quantitative immunoassays run on large-scale immunoassay platforms.

Preparing Revision Guidelines

- Point-by-point responses to the issues raised by the reviewers in a file named "Response to Reviewers," NOT IN YOUR COVER LETTER.
- Upload a compare copy of the manuscript (without figures) as a "Marked-Up Manuscript" file.
- Each figure must be uploaded as a separate file, and any multipanel figures must be assembled into one file.

- Manuscript: A .DOC version of the revised manuscript
- Figures: Editable, high-resolution, individual figure files are required at revision, TIFF or EPS files are preferred

Please return the manuscript within 60 days; if you cannot complete the modification within this time period, please contact me. If you do not wish to modify the manuscript and prefer to submit it to another journal, please notify me of your decision immediately so that the manuscript may be formally withdrawn from consideration by Microbiology Spectrum.

We thank the reviewers for their constructive comments. Below is an item by item response to each comment, in italics. We trust these responses will be in line with expectations.

Reviewer #1 (Comments for the Author):

In this manuscript Dimech et al. perform a qualitative and comparative evaluation of a number of rapid tests (35) used for the detection of SARS-CoV-2 infection. This included 26 rapid diagnostic tests and 8 immunoassays. The manuscript is well written and the data clear. The authors performed an extensive and well-controlled study in which they tested samples expected to be positive, samples expected to be negative and then a number of other samples to account for cross-reactivity and interference. Most of the tests evaluated in this study were actually not performing optimally and did not fulfill the WHO requirements. The strengths of the manuscript rely on the number of tests included in the study, the well-controlled and stringent conditions used, and that the manuscript is also well-balanced thus acknowledging that the study does not evaluate samples from vaccinees or with a past of prior infection. I found the data of value. The study shows how difficult it is to develop a reliable test with great sensitivity and specificity. However, as a main weakness it would have been desirable that the study was done earlier since at this stage of the pandemic it is not so well-timed.

I only have some minor comments:

Line 42: I suggest that the authors clarify here if the individuals were convalescent or currently infected when the samples were taken.

We have added the phrase “previously infected, at underdetermined time periods,” as there was insufficient information regarding the samples pedigree to be definitive”. These samples are quite difficult to acquire.

Line 37: I suggest a different use of the comma, please revise.

Comma deleted

Line 69, 71 and 89: There seem to be links here that are not working, or perhaps it is just the text that was made blue and underlined during the writing process, please correct.

Removed the blue. These were remnants from a previous internal report.

Line 136: please define NAT at first use for enhanced clarity.

The samples were tested in various laboratories throughout Germany and USA. Have added “various commercial NAT” for added clarity.

Line 161: please clarify the difference between icteric samples and the ones with high levels of bilirubin in line 163.

The samples with “high levels of bilirubin” had laboratory-based testing for bilirubin levels, whereas the samples that were “icteric” we selected due to visual features. It was determined that we needed to differentiate these samples accordingly.

Table S2: what is A/B?

Unable to find “A/B” using search functions in either the excel spreadsheet table or in the table heading in the text.

Supplementary materials: I think it would be helpful to add a legend or the like at the bottom of the data.

The legend is in the Table headings which includes "Anti-SARS-CoV-2 IgG, IgM or total antibody results are highlighted as reactive (R), non-reactive (NR) or equivocal (EQ)." On publication, the table and the heading will be merged, so adding an additional legend will result in duplication.

Line 261: out of curiosity, do the authors have a hypothesis or explanation for the samples with rheumatoid factor being the most falsely reactive? Would it be the increased number of autoantibodies?

Rheumatoid factor is known to be cross reactive to a range of other specific antibodies, especially IgM, due to the non-specific nature of antibodies produced. Indeed, in the olden days, we used to do Rf extraction prior to performing testing on a sandwich IgM EIA (as distinct from a reverse capture IgM assay which avoided the problem).

Some typos:

Line 132: there's a parenthesis left behind.

Deleted

Line 342: test should read tests.

Corrected.

Along the text: I would add a hyphen in Hepatitis B e-antigen for clarity.

Not a common way of expressing the analyte but has been changed.

Line 387: wouldn't RDT be rapid diagnostic test and not device?

Changed to rapid diagnostic test

Reviewer #2 (Comments for the Author):

The assessment of performance is sound. However, the authors need to clearly specify in the abstract (and body of the manuscript) that they are reviewing the performance of "rapid tests" for SARS-CoV-2 antibodies. It is not clear that this is the case until you start reading the body of the manuscript.

We have added, in both the abstract and the introduction, that the tests were selected as being suitable for use in low- and middle-income countries. There were not all rapid tests, some being microtitre plate EIAs. This is detailed in Table 1 in column labelled "Test Type".

Rapid tests are pretty much all some variation of disposable lateral flow immunoassay cartridge (or card, or cassette) and hence are all qualitative measures or semi-quantitative at best (although semi-quantitative claims for lateral flow assays are dubious). It would behoove the authors to specify the format of these rapid tests in such a way, i.e., refer to them as lateral flow immunoassays, which is a much more accurate description than calling them ELISAs, which conjures up the thought of a microtiter well plate with the reacting antibodies being read by a spectrophotometer and quantitated against a calibration curve.

Some were microtitre plate EIA and others were lateral flow devices. This is detailed throughout but especially in Table 1.

If there are any rapid assays that deviate from the lateral flow cartridge format, then state that as well, but in my quick review of the methods assessed they all sound like lateral flow immunoassays, or some variation of that.

All RDTs were lateral flow devices.

2.5 years from the start of the pandemic in developed parts of the world use of these rapid assays for clinical purposes is not really happening. Developed countries are, for the most part, all using semi-quantitative or quantitative immunoassays for antibody measurement that require large-scale immunoassay instruments to perform.

Fully agree. The time required to develop a protocol, collect samples and organise budgets, resources and conduct the study, in the middle of an emerging pandemic is fraught. However, we can indicate that the results of the studies were released online as they became available, and a final report was published on the WHO website. This manuscript seeks to create a historical record of the results as publication on websites will become lost to follow up. The reviewer's point does underline the need for well resources organisations to be enabled to conduct highly creditable performance evaluations on emerging infections so evidence-based decisions on test kit selection can be made efficiently.

Thus, it would be appropriate to specify where (what countries) these rapid tests are still being used in, their prevalence of use, and for what purpose (i.e., clinical, research or both).

We have specified that the potential use of these tests are in Low and middle income countries.

The use of these tests is specified in the Discussion pg 372-373 stating "Nonetheless, a small number of products met WHO TPP criteria and in the right context could play a useful role in serosurveillance and epidemiological research." And in the text pg 327-328 stating "These findings would support use of these limited number of RDT and EIA products for serosurveillance, or retrospective diagnosis (in unvaccinated individuals)."

There are some grammatical issues and typos to address (e.g., line 33 "defines" should be "defined", ELISA" is first used on line 40 without definition of the acronym, etc.)

Both corrected

It should also be reinforced throughout the manuscript that these are rapid assays that are being analyzed, i.e., rather than referring to these tests as "serology IVDs" or "IVDs" (e.g., line 29, 30, 81, change to "rapid antibody tests".

Have changed IVD to test kits where it relates to tests evaluated in the study, noting that not all tests evaluated were RDT. Where generally discussing IVD, we have retained the text.

Additionally, the authors seem to separate these rapid tests into two categories, "rapid diagnostic test" or "enzyme immunoassay", but what really matters is whether they were all qualitative assays that were run on disposable cartridges; if so, I would define those as all falling into a "rapid test" category. Again, what constitutes a "rapid test" needs to be specified from the outset, and again, as far as I can tell they are all rapid assays.

Not all test kits evaluated were rapid tests. Some, as specified, were EIAs.

Readers outside laboratory medicine do not understand the differences in assay formats and may conflate this as all antibody IVDs perform similarly to rapid tests. Other antibody IVDs for SARS-CoV-2 being used clinically are CMIA, ECLIA or similar format run on platforms like Abbott, Roche, Diasorin, MSD, Ortho, Siemens, etc., all of which provide semi-quantitative and quantitative measures and are far more robust for other reasons as well (e.g., sensitivity, specificity, etc.). Therefore, I cannot emphasize the importance of clearly defining what types of antibody tests we are talking about, and if these are all in fact rapid qualitative lateral flow immunoassays (or some variation of this), then they should definitely not be conflated with the semi-quantitative and quantitative immunoassays run on large-scale immunoassay platforms.

The changes made have highlighted that the tests evaluated were rapid, lateral flow devices or microtitre plate EIAs. The target analyte is detailed for each in Table 1.

April 14, 2023

Dr. Wayne Dimech
National Reference Laboratory, Australia
4th Floor Healy Building
41 Victoria Parade
Fitzroy, Victoria 3065
Australia

Re: Spectrum05101-22R1 (Comprehensive, Comparative Evaluation of 35 Manual SARS-CoV-2 Serological Assays)

Dear Dr. Wayne Dimech:

Your manuscript has been accepted, and I am forwarding it to the ASM Journals Department for publication. You will be notified when your proofs are ready to be viewed.

Sincerely,

Oliver Laeyendecker
Editor, Microbiology Spectrum
